# Unsupervised feature selection algorithm based on $L_{2,p}$-norm feature reconstruction

**Wei Liu** [1]*, **Qian Ning**[1], **Guangwei Liu**[2], **Haonan Wang**[3], **Yixin Zhu**[1], **Miao Zhong**[1]

**1** College of Science, Liaoning Technical University, Fuxin, Liaoning, China, **2** College of Mines, Liaoning Technical University, Fuxin, Liaoning, China, **3** Johns Hopkins University, Baltimore, Maryland, United States of America

☯ These authors contributed equally to this work.

\* liuwei@lntu.edu.cn

**Data availability statement:** All relevant data are within the paper and its Supporting Information files.

**Funding:** This study was funded through the National Natural Science Foundation of China (52374123) (awarded to GL), the Basic Scientific Research Project of the Liaoning Provincial Department of Education (LJ212410147013) (awarded to WL) and the Basic Scientific Research Project of the Liaoning

## Abstract

Traditional subspace feature selection methods typically rely on a fixed distance to compute residuals between the original and feature reconstruction spaces. However, this approach struggles to adapt to diverse datasets and often fails to handle noise and outliers effectively. In this paper, we propose an unsupervised feature selection method named unsupervised feature selection algorithm based on $l_{2,p}$-norm feature reconstruction (NFRFS). Employing a flexible norm to represent both the original space and the spatial distance of feature reconstruction, enhances adaptability and broadens its applicability by adjusting $p$. Additionally, adaptive graph learning is integrated into the feature selection process to preserve the local geometric structure of the data. Features exhibiting sparsity and low redundancy are selected through the regularization constraint of the inner product in the feature selection matrix. To demonstrate the effectiveness of the method, numerical studies were conducted on 14 benchmark datasets. Our results indicate that the method outperforms 10 unsupervised feature selection algorithms in terms of clustering performance.

## 1. Introduction

In this field of information explosion, traditional data processing methods are facing unprecedented challenges due to the vast amount of data and the high dimensionality. The efficient and accurate processing of these rapidly growing high-dimensional datasets and extracting key information has become a focal point of attention and research in fields such as data mining [1], pattern recognition [2], and machine learning [3]. Feature selection algorithms extract representative features from raw data, not only achieving dimensionality reduction but also preserving the physical significance of the data [4].

Based on whether the data includes label information, feature selection can be divided into three types: supervised, semi-supervised, and unsupervised [5]. Since unsupervised feature selection does not rely on label information, it identifies features that best represent the characteristics of the data by analyzing its intrinsic structure, making it of significant research importance and value [6]. According to evaluation criteria, feature selection methods can be classified into filter, wrapper, and embedded methods [7]. Embedded methods combine the

Provincial Department of Education (LJ212410147019) (awarded to GL). The funders had no role in study design, data collection and analysis, decision to publish, or preparation of the manuscript. There was no additional external funding received for this study.

**Competing interests:** Conflict of interest/Competing interests (check journal-specific guidelines for which heading to use) The authors declare no competing financial interests.

advantages of both filter and wrapper methods, integrating the feature selection process into model training to enhance the performance of algorithms [8].

Graph structure is crucial for feature selection. Authors in [9] introduced the Laplacian Score algorithm, which is based on the relationships between data points. This algorithm evaluates the importance of each feature by calculating its Laplacian score, reflecting its ability to preserve local information. A study in [10] introduced a feature selection algorithm based on latent representation learning and manifold regularization. By combining latent representation learning using non-negative matrix factorization and graph-based manifold regularization, it performs feature selection in a robust latent space, capturing the intrinsic structure of the data and reducing the negative impact of noise. This approach, which relies on a fixed similarity graph and depends on the sample similarity matrix, separates the construction of the graph from the learning of the feature selection matrix. Therefore, it is susceptible to the influence of noise or outliers. Therefore, literature [11] proposed an unsupervised feature selection algorithm based on adaptive structure learning, which simultaneously conducts feature selection and data structure learning to better capture both global and local structures of the data. In literature [12], the unsupervised feature selection method known as self-weighted adaptive graph-based minimum-redundant subspace learning is mentioned. This approach integrates adaptive self-weighted graph learning, minimum redundancy, and sparsity constraints into a comprehensive framework. Manifold regularization can preserve the inherent geometric structure of the data, so unsupervised feature selection algorithms that incorporate manifold regularization typically achieve better performance. Although these algorithms have made some improvements, they need to enhance their handling of redundant information during feature selection.

In recent years, regularizers are often used to constrain the feature selection matrix in dealing with redundant information [13]. A study in [14] combined spectral analysis with $l_1$-norm regularization and proposed the multi-cluster feature selection algorithm. Authors in [15] unified feature selection and similarity matrix construction into a single framework and used an $l_{2,0}$-norm constraint on the feature selection matrix to achieve feature selection. In Reference [16], the authors introduced non-negative constraints and applied an $l_{2,p}$-norm to the matrix of feature transformations. This approach, in comparison to the $l_{2,1}$-norm, offers a more tractable optimization process. The variable $p$ allows for a flexible trade-off between row sparsity and the convexity of the model, potentially enhancing the model performance. Meanwhile, Reference [17] incorporated the absolute values of inner product outcomes between the vectors of the feature selection matrix as a regularization component, thus fully accounting for feature interdependencies in the pursuit of a more independent selection of the subset of features. Influenced by the norms used in the feature selection matrix, regularizers can also be added to the loss function to prevent model overfitting and promote sparsity. Common choices include $F$-norm and $l_{2,1}$-norm, but both assume a fixed distance between the original samples and predicted labels, which limits their ability to flexibly adjust this distance based on the data's structure. Therefore, Reference [18] proposed a feature selection method based on the $l_{2,p}$-norm and sample constraints, applied in the diagnosis of Alzheimer's disease.

Motivated by these considerations, we propose an efficient feature selection technique known as the unsupervised feature selection algorithm based on $l_{2,p}$-norm feature reconstruction (NFRFS). The essence of this algorithm lies in its ability to flexibly adjust the distance between the original data space and the reconstructed subspace through the $l_{2,p}$-norm. Graph embedding technology and feature selection work in tandem to jointly learn the local structural information between data points. This interactive effect enables the algorithm to identify a subset of features that best represent the characteristics of the dataset. By employing inner

product regularization, we further ensure the sparsity of the selected features, while also eliminating redundant or irrelevant features. The effectiveness of this method has been thoroughly demonstrated across various types of datasets. The main contributions of this paper are as follows:

- In the reconstruction error, a more flexible $l_{2,p}$-norm is used to measure the distance between the original samples and the reconstructed samples, and the value of $p$ is adjusted to handle noise and outliers in the dataset.
- The feature selection matrix is sparsified by utilizing the inner product sparse regularization, selecting representative features with low redundancy.
- Comprehensive experiments on 14 benchmark datasets show that NFRFS is superior to several state-of-the-art feature selection methods. The experimental results validate the effectiveness and practicality of the model.

The rest of the paper is organized as follows. Sect 2 explains some basic notions and definitions. In Sect 3, we propose an optimization problem for feature selection and an iterative algorithm for solving the problem. In Sect 4, various experimental results are analyzed. Conclusions are drawn in Sect 5.

## 2. Related work

### 2.1. Notation and definition

To help explain the details of the proposed algorithm, some notations need to be introduced in advance. For a matrix $X \in \mathbb{R}^{n \times d}$, its $l_{r,s}$-norm can be defined as follows:

$$\|X\|_{r,s} = \left( \sum_{i=1}^{d} \left( \sum_{j=1}^{n} x_{ij}^{r} \right)^{\frac{s}{r}} \right)^{\frac{1}{s}}$$

The norm is called the $F$-norm or the $l_2$-norm when $r = s = 2$. Based on the above definition of the norm, the $F$-norm and $l_{2,p}$-norm of matrix are calculated as

$$\|X\|_F = \left( \sum_{i=1}^{d} \sum_{j=1}^{n} x_{ij}^{2} \right)^{\frac{1}{2}}$$

$$\|X\|_{2,p} = \left( \sum_{i=1}^{d} \left( \sum_{j=1}^{n} x_{ij}^{2} \right)^{\frac{p}{2}} \right)^{\frac{1}{p}}$$

More details are listed in Table 1.

### 2.2. Feature reconstruction in subspace using $l_{2,p}$-norm

Facing the challenges of high dimensionality and excessive redundant information, Wang [19] proposed an algorithm from the perspective of subspace learning. The algorithm extracts a low-dimensional subspace from high-dimensional data, which can represent the main information of the original feature space while removing redundancy and noise.

$$\begin{aligned} & \text{argmin} \, \|X - XWH\|_F^2 \\ & s.t. W.H \geq 0, \, W^T W = I_l \end{aligned} \tag{1}$$

**Table 1. The notations used in this paper.**

| Notation | Description |
|---|---|
| $n$ | Number of samples |
| $d$ | Original spatial dimension |
| $l$ | The number of selected features |
| $x^i$ | The $i$-th row of $X$ |
| $x_j$ | The $j$-th column of $X$ |
| $x_{ij}$ | The element in $i$-th row and $j$-th column of $X$ |
| $s_{ij}$ | The similarity between the sample points $x_i$ and $x_j$ |
| $X \in \mathbb{R}^{n \times d}$ | Original data matrix |
| $W \in \mathbb{R}^{d \times l}$ | The feature selection matrix |
| $H \in \mathbb{R}^{l \times d}$ | The reconstruction coefficient matrix |
| $I \in \mathbb{R}^{l \times l}$ | The identity matrix |
| $L \in \mathbb{R}^{n \times n}$ | The Laplacian matrix |
| $S \in \mathbb{R}^{n \times n}$ | The similarity matrix of data space |
| $D \in \mathbb{R}^{n \times n}$ | Diagonal matrix |
| $X^T$ | The transpose of $X$ |
| $tr(X)$ | The trace of $X$ |
| $\langle a, b \rangle$ | The inner product of $a$ and $b$ |

Where $H \in \mathbb{R}^{l \times d}$ is the coefficient matrix used for reconstruction, which maps the learned subspace to the original space. $l$ represents the number of selected features, and $I \in \mathbb{R}^{l \times l}$ is an identity matrix. $W \in \mathbb{R}^{d \times l}$ is the feature selection matrix, constrained with orthogonality $W$ to ensure that there is at most one non-zero value per row and column. Additionally, non-negative constraints are imposed on $W$ to preserve its real-world physical meaning [20].

In existing subspace feature selection methods, the $F$-norm is typically used to measure the distance between the original data space and the reconstructed subspace [21]. However, for some datasets, using a fixed distance metric does not result in the optimal feature subset. Therefore, this paper uses an adaptive distance metric to effectively improve model performance, choosing the $l_{2,p}$-norm to constrain the distance between the original space and the reconstructed subspace. The $l_{2,p}$-norm allows for flexible adjustment of the size of parameter $p$, choosing the $p$ parameter most favorable for feature selection. The model's application of the $l_{2,p}$-norm can be expressed as:

$$\operatorname*{argmin} \|X - XWH\|_F^2$$
$$s.t. W.H \geq 0, W^T W = I_l$$

(2)

Choosing different $p$ values significantly impacts the model's performance. Fig 1 presents 3D surface plots for three different norms, showing that both $F$-norm and $l_{2,1}$-norm tend to optimize more towards the origin. However, during the optimization process, the $l_{2,1/2}$-norm is more inclined towards the coordinate axes. Therefore, using the $l_{2,1/2}$-norm to constrain the model effectively eliminates redundant features and selects more discriminative features.

## 2.3. Sparsity regularization term of the feature selection matrix

Reference [22] emphasizes the critical role of the feature weight matrix $W$ in the feature selection process and highlights the necessity of introducing appropriate regularization techniques for effectively learning $W$. However, conventional regularization methods each have certain limitations: Although $l_1$-norm regularization can achieve sparsity, it can easily lead to underfitting in high-dimensional data [23]; while $l_{2,0}$-norm regularization can produce desirable

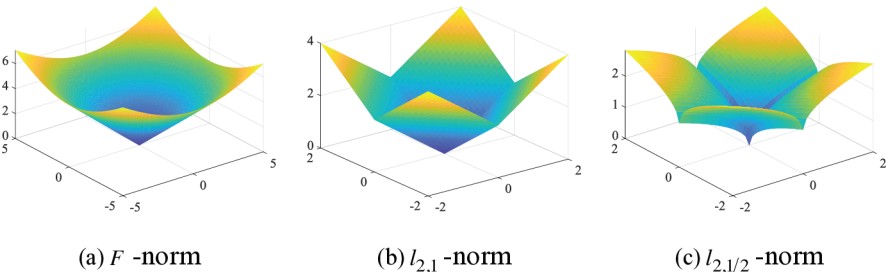

(a) $F$ -norm (b) $l_{2,1}$ -norm (c) $l_{2,1/2}$ -norm

**Fig 1. D surface plot of $F$-norm, $l_{2,1}$-norm and $l_{2,1/2}$-norm regularization term.**

sparsity, its non-convexity and non-smooth nature make the optimization process extremely challenging [24]. Moreover, $l_{2,1}$-norm regularization often overlooks correlations among features, resulting in limited performance improvements when dealing with highly redundant features in high-dimensional data.

To address these issues, reference [25] proposed an inner product-based regularization method. By continually reducing the inner products between feature vectors during optimization, this approach forces them closer to zero, thereby effectively mitigating feature redundancy. Several empirical studies have confirmed both the efficacy and superiority of the inner product regularization term. For instance, studies reported in references [26,27] have demonstrated that features selected by the inner product regularization exhibit significantly lower inter-feature correlations compared to those selected by $l_{2,1}$-norm regularization. Another study [25] showed that incorporating the inner product regularization term into the feature selection process not only outperforms traditional regularization techniques in terms of clustering performance but also achieves exceptional results with fewer selected features. Collectively, these findings indicate that the inner product regularization term helps select more representative and less redundant features, thereby enhancing the overall performance of feature selection. The inner product regularization term for the feature selection matrix $W$ is defined as:

$$\sum_{i,j=1,i\neq j}^{d} < w^i, w^j >= \sum_{i,j=1,i\neq j}^{d} w^i w^{j^T} = Tr(1_{d\times d}WW^T) - Tr(WW^T) \tag{3}$$

## 2.4. Adaptive graph learning

In the design of existing model, the global structure and constraints on the feature selection matrix have been thoroughly considered. To further enhance the generalizability of the model, local structural information is incorporated, and the most effective way to achieve this is by introducing graph learning [28]. Graph structures can effectively preserve the local neighborhood information of data, and when mapping from the original feature space to a lower-dimensional feature space, they can maintain the geometric structure of the samples [29]. Simply put, if two sample points are close to each other in the original space, they should also remain close in the feature-selected projection space. Mathematically, this can be expressed as:

$$\min_{S} \sum_{i,j=1}^{n} ||W^T x_i - W^T x_j||_2^2 s_{ij} = Tr(W^T X^T LXW) \tag{4}$$

However, real-world data is often affected by noise, which making the *k*-nearest neighbor graph constructed using the aforementioned methods susceptible to inaccuracies. To address this issue, researchers in [30] have proposed adaptive graph learning as an effective solution to the problems of imbalanced neighbors and feature redundancy. Adaptive graph learning dynamically computes the similarity matrix during the optimization process, enabling it to more accurately capture the true relationships between samples without relying on predefined Euclidean or cosine distances. By dynamically adjusting neighbor relationships, adaptive graph learning overcomes the limitations of traditional predefined methods and significantly enhances the model's robustness to noise and adaptability to high-dimensional data.

$$\min_{S} Tr(W^T X^T L X W) + \gamma ||S||_F^2$$
$$s.t. \sum_{j=1}^{n} s_{ij} = 1, s_{ij} \geq 0 \tag{5}$$

Where $L$ is the Laplacian matrix, and $L = D - S$. $S \in R^{n \times n}$ represents the similarity matrix of the samples, where the element $s_{ij}$ denotes the similarity between the sample points $x_i$ and $x_j$. The matrix $S$ can be calculated adaptively, $\gamma$ influences the nearest neighbor number of each sample. $D \in R^{n \times n}$ is a diagonal matrix and its diagonal elements are defined as:

$$d_{ii} = \sum_{j=1}^{n} s_{ij} \tag{6}$$

## 3. Unsupervised feature selection based on $l_{2,p}$-norm feature reconstruction

### 3.1. The objective function of NFRFS

In constructing the subspace, the $l_{2,p}$-norm is used to flexibly measure the distance between the original samples and the reconstructed samples. The adaptive graph embedding learning takes into account the similarity relationships between samples, preserving the local geometric structure of the data. In addition, by applying an inner product constraint on the feature selection matrix, a more sparse solution can be obtained to help to select a representative subset of features. The final objective function is expressed as:

$$\min ||X - XWH||_{2,p}^p + \alpha Tr(W^T X^T L X W) +$$
$$\beta (Tr(1_{d \times d} WW^T) - Tr(WW^T)) + \gamma ||S||_F^2$$
$$s.t. W \geq 0, H \geq 0, W^T W = I, \sum_{j=1}^{n} s_{ij} = 1, s_{ij} \geq 0 \tag{7}$$

Where $\alpha$ and $\beta$ are regularization parameters, and $\gamma$ is a coefficient that can be determined during the optimization process.

### 3.2. Optimization

The objective function in Eq (7) includes three variables, *W*, *H* and *S*. To improve computational efficiency, this paper employs an alternate optimization method to optimize

the objective function, that is, by fixing two variables each time and optimizing the other variable.

Define two Lagrange multipliers, $\theta$ and $\mu$, to ensure the non-negativity of the matrices $W$ and $H$. The resulting Lagrangian function is as follows:

$$
\begin{aligned}
L(W,H) = & \|X - XWH\|_{2,p}^p + \alpha Tr(W^T X^T LXW) \\
& + \beta(Tr(1_{d\times d}WW^T) - Tr(WW^T)) \\
& + \frac{\lambda}{2}(W^T W - I) + Tr(\theta W^T) + Tr(\mu H^T)
\end{aligned}
\tag{8}
$$

Define a diagonal matrix $U$, with diagonal elements being $u_{ii} = \frac{p}{2\|(X-XWH)_i\|_2^{2-p}}$.

1. Fix $H$, $S$ and Update $W$: By taking the partial derivative of Eq (8) with respect to $W$, the following formula can be obtained:

$$
\begin{aligned}
\frac{\partial L}{\partial W} = & -X^T UXH^T + X^T UXWHH^T + \alpha X^T LXW + \beta(WW^T W - W) \\
& + \lambda(1_{d\times d}W - W) + \theta
\end{aligned}
\tag{9}
$$

By using the Karush–Kuhn–Tucker (KKT) conditions $\theta_{ij} W_{ij} = 0$, the obtained formula is as follows:

$$
\begin{aligned}
(-X^T UXH^T + X^T UXWHH^T + \alpha X^T LXW \\
+ \beta(WW^T W - W) + \lambda(1_{d\times d}W - W))_{ij} W_{ij} = 0
\end{aligned}
\tag{10}
$$

Thus, the update rule for $W$ is as follows:

$$
W_{ij} \leftarrow W_{ij} \frac{[XUX^T H^T + \alpha X^T SXW + \beta W + \lambda W]_{ij}}{[XUX^T WHH^T + \alpha X^T DXW + \beta WW^T W + \lambda 1_{d\times d}W]_{ij}}
\tag{11}
$$

2. Fix $W$, $S$ and Update $H$: By taking the partial derivative of Eq (8) with respect to $H$, the following formula can be obtained:

$$
\frac{\partial L}{\partial H} = -W^T X^T UX + W^T X^T UXWH^T + \mu
\tag{12}
$$

By using the Karush–Kuhn–Tucker (KKT) conditions $\mu_{ij} H_{ij} = 0$, the obtained formula is as follows:

$$
(-W^T X^T UX + W^T X^T UXWH^T)_{ij} H_{ij} = 0
\tag{13}
$$

Thus, the update rule for $H$ is as follows:

$$
H_{ij} \leftarrow H_{ij} \frac{[W^T X^T UX]_{ij}}{[W^T X^T UXWH^T]_{ij}}
\tag{14}
$$

3. Fix $W$, $H$ and Update $S$: By taking the partial derivative of Eq (7) with respect to $S$, the following formula can be obtained:

$$\min_{S} \sum_{i,j=1}^{n} (\alpha ||W^T x_i - W^T x_j||_2^2 s_{ij} + \gamma s_{ij}^2)$$

$$s.t. \sum_{j=1}^{n} s_{ij} = 1, s_{ij} \geq 0 \tag{15}$$

Denote $d_{ij} = ||W^T x_i - W^T x_j||_2^2$, so we can transform Eq (15) to a vector form as

$$\min_{s_i^T \mathbf{1}=1, s_i \geq 0} ||s_i + \frac{\alpha}{2\gamma} d_i||_2^2$$

$$s.t. \sum_{j=1}^{n} s_{ij} = 1, s_{ij} \geq 0 \tag{16}$$

Introduce Lagrange multipliers $\omega$ and $\varphi_i$ to construct the Lagrangian function:

$$\mathcal{L}(s_i, \omega, \varphi_i) = ||s_i + \frac{\alpha}{2\gamma} d_i||_2^2 - \omega(s_i^T \mathbf{1} - 1) - \varphi_i^T s_i \tag{17}$$

According to the KKT conditions, the optimal solution is obtained.

$$s_{ij} = (-\frac{\alpha d_{ij}}{2\gamma_i} + \omega)_+ \tag{18}$$

In unsupervised feature selection algorithms, preserving the local geometric manifold structure of the data tends to be more effective than preserving the global structure. Therefore, only neighboring points $k$ are considered to construct the similarity matrix. The optimal solution for $\gamma$ can be represented as the average of all $\gamma_i$ [31]. Assuming $d_{i1}, d_{i2}, ..., d_{in}$ is sorted from smallest to largest and satisfies the condition.Because $s_i$ satisfies $s_{ik} > 0 \geq s_{i,k+1}$, we have then we have:

$$\begin{cases} s_{ik} > 0 \Rightarrow -\frac{\alpha d_{ik}}{2\gamma_i} + \omega > 0 \\ s_{i,k+1} \leq 0 \Rightarrow -\frac{\alpha d_{i,k+1}}{2\gamma_i} + \omega \leq 0. \end{cases} \tag{19}$$

According to Eq (18) and the constraint $s_i^T 1 = 1$ we have

$$\sum_{j=1}^{k} (-\frac{\alpha d_{ij}}{2\gamma_i} + \omega) = 1 \Rightarrow \omega = \frac{1}{k} + \frac{\alpha}{2k\gamma_i} \sum_{i=1}^{k} d_{ij} \tag{20}$$

By substituting the value of $\omega$ in Eq (19) into Eq (18), we have

$$\frac{\alpha}{2} \left( k d_{ik} - \sum_{j=1}^{k} d_{ij} \right) < \gamma_i \leq \frac{\alpha}{2} \left( k d_{i,k+1} - \sum_{j=1}^{k} d_{ij} \right) \tag{21}$$

Therefore, in order to obtain an optimal solution of $s_i$ that has exact $k$ nonzero values, we set $\gamma_i$ to be

$$\gamma_i = \frac{\alpha}{2} \left( k d_{i,k+1} - \sum_{j=1}^{k} d_{ij} \right) \tag{22}$$

and then the overall $\gamma$ is set to the mean of $\gamma_i$ as

$$\gamma = \frac{1}{n}\sum_{i=1}^{n}\left(\frac{\alpha k}{2}d_{i,k+1} - \frac{\alpha}{2}\sum_{j=1}^{k}d_{ij}\right) \tag{23}$$

Finally, substitute Eq (22) into Eq (18), and consider only $k$ neighboring points to construct the similarity matrix. In summary, the solution can be obtained by solving as

$$s_{ij} = \begin{cases} \dfrac{d_{i,k+1}-d_{ij}}{\alpha k d_{i,k+1}-\alpha\sum_{j=1}^{k}d_{ij}}, j \leq k \\ 0, j \geq k+1 \end{cases} \tag{24}$$

**Algorithm 1:** NFRFS
**Input:** Date matrix $X$; parameter $\alpha$, $\beta$, $p$; select feature number $l$; gaussian scale parameter $\sigma$.
**Output:** Calculate and sort $\|w_i\|_2$ in the descending order, then select the top $m$ ranked features as the results of feature selection.
1: Initialize $t = 1$, $W$, $H$, maxIter
2: while $t \leq$ maxIter do
3: update $W$ by Eq (11)
4: update $H$ by Eq (14)
5: update $S$ by Eq (24)
6: **end while**

## 3.3. Convergence analysis

To prove the convergence of Algorithm 1 for updating the variable $W$, the following the lemma is introduced [32]:

**Lemma 1:** Assume $g_i^t$ and $g_i^{t+1}$ as the $i$-th row of matrices $G_t$ and $G_{t+1}$, respectively, thus for $p \in (0,1]$ we have:

$$\|g_i^{t+1}\|_2^p - \frac{p}{2}\left(\frac{\|g_i^{t+1}\|_2^2}{\|g_i^t\|_2^{2-p}}\right) \leq \|g_i^t\|_2^p - \frac{p}{2}\left(\frac{\|g_i^t\|_2^2}{\|g_i^t\|_2^{2-p}}\right) \tag{25}$$

$$\Rightarrow \sum_{i=1}^{r}\left(\|g_i^{t+1}\|_2^p - \frac{p}{2}\left(\frac{\|g_i^{t+1}\|_2^2}{\|g_i^t\|_2^{2-p}}\right)\right) \leq \sum_{i=1}^{r}\left(\|g_i^t\|_2^p - \frac{p}{2}\left(\frac{\|g_i^t\|_2^2}{\|g_i^t\|_2^{2-p}}\right)\right) \tag{26}$$

**Proof.** For convenience in writing, we have Eq (27) holds.

$$\begin{aligned} F(W) = &\alpha \mathrm{Tr}\left((W^T X^T L X W)\right) \\ &+ \beta\left(\mathrm{Tr}\left(1_{d\times d}W(W)^T\right) - \mathrm{Tr}\left(W^{(t)}(W^{(t)})^T\right)\right) \\ &+ \frac{\lambda}{2}\left((W^T W - I)\right) \end{aligned} \tag{27}$$

According to step 3 in Algorithm 1, the following inequality holds.

$$
\begin{aligned}
&Tr\left((X - XW^{(t+1)}H^{(t)})^T U^{(t)}(X - XW^{(t+1)}H^{(t)})\right) + F(W^{(t+1)}) \\
&\leq \mathrm{Tr}\left((X - XW^{(t)}H^{(t)})^T U^{(t)}(X - XW^{(t)}H^{(t)})\right) + F(W^{(t)})
\end{aligned}
\tag{28}
$$

Meanwhile, the following two equations hold.

$$
\begin{aligned}
&Tr\{(X - XW^{(t)}H^{(t)})^T U^{(t)}(X - XW^{(t)}H^{(t)})\} \\
&= \sum_i \frac{p}{2} \frac{\|(X - XW^{(t)}H^{(t)})_i\|_2^2}{\|(X - XW^{(t)}H^{(t)})_i\|_2^{2-p}}
\end{aligned}
\tag{29}
$$

$$
\begin{aligned}
&Tr\{(X - XW^{(t+1)}H^{(t)})^T U^{(t)}(X - XW^{(t+1)}H^{(t)})\} \\
&= \sum_i \frac{p}{2} \frac{\|(X - XW^{(t+1)}H^{(t)})_i\|_2^2}{\|(X - XW^{(t)}H^{(t)})_i\|_2^{2-p}}
\end{aligned}
\tag{30}
$$

Therefore, it is easy to get

$$
\begin{aligned}
&\sum_i \frac{p}{2} \frac{\|(X - XW^{(t+1)}H^{(t)})_i\|_2^2}{\|(X - XW^{(t)}H^{(t)})_i\|_2^{2-p}} + F(W^{(t+1)}) \\
&\leq \sum_i \frac{p}{2} \frac{\|(X - XW^{(t)}H^{(t)})_i\|_2^2}{\|(X - XW^{(t)}H^{(t)})_i\|_2^{2-p}} + F(W^{(t)})
\end{aligned}
\tag{31}
$$

$$
\begin{aligned}
\Rightarrow & \sum_i \frac{p}{2} \frac{\|(X - XW^{(t+1)}H^{(t)})_i\|_2^2}{\|(X - XW^{(t)}H^{(t)})_i\|_2^{2-p}} + F(W^{(t+1)}) \\
& - \sum_i \|(X - XW^{(t+1)}H^{(t)})_i\|_2^p + \sum_i \|(X - XW^{(t+1)}H^{(t)})_i\|_2^p \\
\leq & \sum_i \frac{p}{2} \frac{\|(X - XW^{(t)}H^{(t)})_i\|_2^2}{\|(X - XW^{(t)}H^{(t)})_i\|_2^{2-p}} + F(W^{(t)}) \\
& - \sum_i \|((X - XW^{(t)}H^{(t)})_i\|_2^p + \sum_i \|(X - XW^{(t)}H^{(t)})_i\|_2^p
\end{aligned}
\tag{32}
$$

$$
\begin{aligned}
\Rightarrow & F(W^{(t+1)}) + \sum_i \|(X - XW^{(t+1)}H^{(t)})_i\|_2^p \\
& - \left\{ \sum_i \|(X - XW^{(t+1)}H^{(t)})_i\|_2^p - \sum_i \frac{p}{2} \frac{\|(X - XW^{(t+1)}H^{(t)})_i\|_2^2}{\|(X - XW^{(t)}H^{(t)})_i\|_2^{2-p}} \right\} \\
\leq & F(W^{(t)}) + \sum_i \|(X - XW^{(t)}H^{(t)})_i\|_2^p \\
& - \left\{ \sum_i \|(X - XW^{(t)}H^{(t)})_i\|_2^p - \sum_i \frac{p}{2} \frac{\|(X - XW^{(t)}H^{(t)})_i\|_2^2}{\|(X - XW^{(t)}H^{(t)})_i\|_2^{2-p}} \right\}
\end{aligned}
\tag{33}
$$

It is proved in Lemma 1 that

$$
\begin{aligned}
&\sum_i \left\{ \|(X - XW^{(t+1)}H^{(t)})_i\|_2^p - \frac{p}{2}\frac{\|(X - XW^{(t+1)}H^{(t)})_i\|_2^2}{\|(X - XW^{(t)}H^{(t)})_i\|_2^{2-p}} \right\} \\
&\leq \sum_i \left\{ \|(X - XW^{(t)}H^{(t)})_i\|_2^p - \frac{p}{2}\frac{\|(X - XW^{(t)}H^{(t)})_i\|_2^2}{\|(X - XW^{(t)}H^{(t)})_i\|_2^{2-p}} \right\}
\end{aligned}
\tag{34}
$$

By summing the Eqs (33) and (34), we arrive at

$$
\begin{aligned}
&\sum_i \|(X - XW^{(t+1)}H^{(t)})_i\|_2^p + F(W^{(t+1)}) \\
&\leq \sum_i \|(X - XW^{(t)}H^{(t)})_i\|_2^p + F(W^{(t)})
\end{aligned}
\tag{35}
$$

$$
\begin{aligned}
\Rightarrow &\|X - XW^{(t+1)}H^{(t)}\|_{2,p}^p + \alpha\,\mathrm{Tr}\left((W^{(t+1)})^T X^T LXW^{(t+1)}\right) \\
&+ \beta\left(\mathrm{Tr}\left(1_{d\times d}W^{(t+1)}(W^{(t+1)})^T\right) - \mathrm{Tr}\left(W^{(t+1)}(W^{(t+1)})^T\right)\right) \\
&+ \frac{\lambda}{2}((W^{(t+1)})^T W^{(t+1)} - I) \\
\leq &\|X - XW^{(t)}H^{(t)}\|_{2,p}^p + \alpha\,\mathrm{Tr}\left((W^{(t)})^T X^T LXW^{(t)}\right) \\
&+ \beta\left(\mathrm{Tr}\left(1_{d\times d}W^{(t)}(W^{(t)})^T\right) - \mathrm{Tr}\left(W^{(t)}(W^{(t)})^T\right)\right) \\
&+ \frac{\lambda}{2}((W^{(t)})^T W^{(t)} - I)
\end{aligned}
\tag{36}
$$

Hence, we can conclude that in each iteration, the value of the objective function concerning $W$ in Eq (11) can monotonically decrease through the update rule of $W$.

A similar proof process can be used to demonstrate that the update rule for variable $H$ also guarantees that the objective function does not increase. Due to the closed-form expression of $S$, when $S$ is optimized while keeping $W$ and $H$ fixed, it is non-increasing. Therefore, when all variables are updated, the value of the corresponding objective function will decrease.

### 3.4. Computational complexity analysis

The complexity of Algorithm 1, including both time complexity and space complexity, will be analyzed in this section. Let $n$ represent the total number of samples, $d$ represent the number of sample features, $l$ represent the number of selected features, and $t$ represent the maximum number of iterations. According to Eq (11), the time complexity for updating matrix $W$ is $O(n^2 d + nd + dl)$, according to Eq (14), the time complexity for updating matrix $H$ is $O(nd + d^2 n)$, and according to Eq (24), the time complexity for updating matrix $S$ is $O(n^2)$. Since the number of selected features is less than the number of samples and the dimensionality of the data, the overall time complexity of the algorithm is $O(t(n^2 d + d^2 n))$. Throughout the algorithm, we need to store data and related variables $O(nd + dl)$. The space complexity of constructing the similarity matrix $S$ based on adaptive graph learning is $O(n^2)$. So the space complexity of the algorithm is $O(n^2 + nd)$.

Table 2 visually lists the time complexity of the comparison algorithm and the time complexity of our algorithm.

**Table 2. Computational complexity of all methods.**

| Methods | Computational complexity |
|---------|--------------------------|
| LS | $O(n^2 d)$ |
| MCFS | $O(d^3 + n^2 m + d^2 n)$ |
| SPFS | $O(n^2 d)$ |
| VSCDFS | $O(d^2)$ |
| AUFS | $O(d^3)$ |
| GLUFS | $O(max(n^3, d^3))$ |
| HSL | $O(d^3 + ndm + 1)$ |
| LRPFS | $O(dn^2 + nd^2)$ |
| RAFG | $O(d^3 + n^3 + n^2 c + ndc)$ |
| NFRFS | $O(n^2 d + d^2 n)$ |

## 4. Experiment

In this section, we conduct experiments to demonstrate the effectiveness and superiority of our proposed NFRFS. All experiments were implemented using Matlab 2022a programming environment in a machine having 2.1 GHz Intel(R) Core(TM) i7-12700F CPU and 16 GB RAM.

### 4.1. Datasets

We evaluate the effectiveness of the feature selection model on 14 datasets, spanning five different fields, including six face image datasets (Yale, warpPIE10P, warpAR10P, ORL, JAFFE, ATT40), three biological datasets (lung, TOX-171, Lung_small), one object image dataset (COIL20), one speech signal dataset (Isolet), and two text datasets (PCMAC, RELATHE). Details about the number of samples, features, classes, types, and sources are provided in Table 3.

### 4.2. Comparison methods

To validate the effectiveness of the proposed approach in unsupervised feature selection, the proposed approach is compared with a baseline method that performs clustering with all the original features and nine other representative existing unsupervised feature selection methods.

LS [9]: Laplacian Score (LS) uses the local geometric information of data to select features, and calculates the score of each feature separately.

MCFS [14]: Multi-cluster feature selection (MCFS) is a multi-cluster feature selection method that initially performs spectral analysis followed by feature selection via sparse regression.

SPFS [36]: Structured learning for unsupervised feature selection with high-order matrix factorization (SPFS) integrates local and global structures into a unified framework and formulates the framework as a form of high-order matrix decomposition.

VSCDFS [37]: Unsupervised feature selection based on variance-covariance subspace distance (VSCDFS) selects a representative feature subset using variance-covariance information of the feature space.

AUFS [38]: Adaptive unsupervised feature selection with robust graph regularization (AUFS) performs unsupervised feature selection by minimizing an objective function that includes self-representation reconstruction error, $l_{2,p}$-norm regularization term, and robust graph regularization term.

**Table 3. Detail introduction to datasets.**

| Dataset | Samples | Features | Classes | Type | Source |
|---------|---------|----------|---------|------|--------|
| Yale | 165 | 1024 | 15 | Face Image | http://www.cad.zju.edu.cn/home/dengcai/Data/data.html |
| lung | 203 | 3312 | 5 | Biological | https://jundongl.github.io/scikit-feature/datasets.html |
| COIL20 | 1440 | 1024 | 20 | Object Image | http://www.cad.zju.edu.cn/home/dengcai/Data/data.html |
| warpPIE10P | 210 | 2420 | 10 | Face Image | https://jundongl.github.io/scikit-feature/datasets.html |
| warpAR10P | 130 | 2400 | 10 | Face Image | https://jundongl.github.io/scikit-feature/datasets.html |
| ORL | 400 | 1024 | 40 | Face Image | https://jundongl.github.io/scikit-feature/datasets.html |
| JAFFE | 213 | 256 | 10 | Face Image | [33] |
| ATT40 | 400 | 1024 | 40 | Face Image | [34] |
| TOX-171 | 171 | 5748 | 4 | Biological | https://jundongl.github.io/scikit-feature/datasets.html |
| Isolet | 1560 | 617 | 26 | Speech signal | https://jundongl.github.io/scikit-feature/datasets.html |
| binalpha | 1404 | 320 | 36 | Handwritten Digit | [35] |
| Lung_small | 73 | 325 | 7 | Biological | https://jundongl.github.io/scikit-feature/datasets.html |
| PCMAC | 1943 | 3289 | 2 | Text | https://jundongl.github.io/scikit-feature/datasets.html |
| RELATHE | 1427 | 4322 | 2 | Text | https://jundongl.github.io/scikit-feature/datasets.html |

GLUFS [39]: Unsupervised feature selection through combining graph learning and $l_{2,0}$-norm constraint (GLUFS) integrates the construction of similarity matrices and feature selection into a unified framework, introducing a sparse learning strategy with $l_{2,0}$-norm constraint.

HSL [40]: Unsupervised feature selection with high-order similarity learning (HSL) simultaneously learns the projection matrix, first-order similarity information, and higher-order similarity information within a unified framework.

LRPFS [41]: Unsupervised Feature Selection with Latent Relationship Penalty Term (LRPFS) explicitly assigns attribute scores to each sample based on its unique importance in the clustering results.

RAFG [42]: Adaptive and flexible $l_1$-norm graph embedding for unsupervised feature selection (RAFG) incorporates the $l_{2,1}$-norm into the elastic regression term and characterizes clustering distributions through adaptive $l_1$-norm graph learning with consistent embeddings.

### 4.3. Evaluation metrics

To verify the clustering performance of the algorithm, this article adopts two evaluation indicators, namely cluster accuracy (ACC) [43] and Normalized Mutual Information (NMI) [44]. Both values are within the range of [0, 1]. The higher the values of ACC and NMI, the better the clustering effect and the more representative the selected feature subset.

1. ACC

$$ACC = \frac{1}{n} \sum_{i=1}^{n} \delta(w_i, map(m_i))$$

Where $w_i$ denotes the ideal label, $m_i$ represents the predicted label, $map(\cdot)$ denotes the optimal mapping function, and $\delta(\cdot)$ represents the indicator function. If $a = b$, then $\delta(a,b) = 1$, otherwise $\delta(a,b) = 0$.

2. NMI

$$NMI = \frac{I(P;Q)}{\sqrt{H(P)H(Q)}}$$

Where $P$ and $Q$ denote the clustering label and the real label, $H(P)$ and $H(Q)$ are the entropies of $P$ and $Q$, and $I(P;Q)$ is the mutual information between $P$ and $Q$. NMI ranges from 0 to 1. NMI is 1 when the two sets are identical and 0 when they are independent. In general, higher ACC values and higher NMI values indicate better performance.

## 4.4. Experimental settings

It is necessary to specify several parameter values for the NFRFS method proposed in this paper and other comparison methods before starting the experiment. The regularization parameters of all algorithms are set within the range $\{10^{-6}, 10^{-4}, 10^{-2}, 1, 10^2, 10^4, 10^6\}$ using the grid search method, and the best results of all algorithms are recorded. The feature selection range of all algorithms is set to $\{20, 40, 60, \cdots, 160, 180, 200\}$. The K-means algorithm is used to cluster the data points formed by the selected features to evaluate different methods. Considering that the K-means algorithm is sensitive to initialization, the experiment repeats the K-means algorithm 20 times to eliminate the influence of initial points on the clustering effect.

## 4.5. Results and analysis

Tables 4 and 5 respectively present the best clustering accuracy (ACC) and normalized mutual information (NMI) scores achieved by NFRFS and other comparative algorithms on 14 datasets, along with the corresponding standard deviations (STD). The highest values among different algorithms for the same dataset are highlighted in bold black font in the tables. The specific results are shown in the tables.

As can be seen from Tables 4 and 5, within the 14 datasets comprising six diverse data types, the ACC and NMI of the NFRFS algorithm significantly exceed those of the baselines using all the original features. This indicates that the NFRFS algorithm is capable of achieving superior ACC and NMI with a relatively smaller number of selected features, which effectively validates the efficacy of the NFRFS algorithm in enhancing clustering performance.

Across the 14 datasets tested, NFRFS achieves the highest ACC and NMI scores on all but three datasets (COIL20, binalpha, and RELATHE). However, for the COIL20, binalpha, and RELATHE datasets, although the clustering performance of NFRFS is slightly inferior, it is very close to the best clustering results achieved by other methods. On the COIL20 dataset, the ACC of the NFRFS algorithm is marginally lower than that of AUFS and HSL. For the binalpha dataset, both the ACC and NMI of NFRFS acquire sub-optimal values. Notably, on the RELATHE dataset, the NMI performance of NFRFS remains higher than that of all other algorithms.

It can be observed from Table 4 that the improvement achieved by NFRFS on the warp-PIE10P dataset is the most remarkable. Specifically, it has increased by 4.26% compared to the second-best algorithm and by 32.5% compared to the method that employs all features for clustering. As shown in Table 5, the enhancement of NFRFS on the PCMAC dataset is the most prominent. The NMI of the other eleven algorithms is all below 10%, while the performance of NFRFS reaches 12.18%. Compared with the SPFS algorithm, the proposed NFRFS

**Table 4. Best ACC for different methods on different datasets (mean±std%).**

| Dataset | Baseline | LS | MCFS | SPFS | VSCDFS | AUFS | GLUFS | HSL | LRPFS | RAFG | NFRFS |
|---|---|---|---|---|---|---|---|---|---|---|---|
| Yale | 38.64±3.61 | 36.94±2.17 | 40.61±3.33 | 39.36±2.36 | 37.12±2.28 | 46.70±3.14 | 45.97±3.90 | 49.70±3.43 | 38.97±2.15 | 50.03±3.61 | **50.39±3.18** |
| lung | 72.46±10.20 | 57.91±7.48 | 72.02±7.57 | 75.37±6.71 | 60.47±7.43 | 79.68±4.52 | 81.55±10.03 | 79.85±3.94 | 63.87±1.40 | 77.64±3.29 | **84.04±4.14** |
| COIL20 | 59.17±3.98 | 53.89±3.34 | 60.92±3.52 | 68.27±2.24 | 60.56±4.41 | 70.26±1.82 | 68.68±2.86 | **71.05±3.32** | 68.91±3.26 | 62.14±2.40 | 69.94±2.25 |
| warpPIE10P | 26.24±2.03 | 44.71±3.00 | 28.86±2.56 | 48.52±3.27 | 25.79±1.21 | 54.48±1.26 | 37.21±2.76 | 45.90±0.99 | 34.17±1.53 | 52.45±1.96 | **58.74±3.96** |
| warpAR10P | 23.58±3.94 | 33.08±3.08 | 30.54±3.23 | 48.92±2.86 | 26.88±3.39 | 40.92±1.72 | 40.23±2.64 | 43.15±3.30 | 43.81±2.97 | 42.65±2.69 | **51.62±3.51** |
| ORL | 51.79±3.37 | 40.09±2.20 | 52.71±3.02 | 51.01±2.26 | 48.21±2.97 | 53.70±1.92 | 54.43±2.90 | 54.89±2.24 | 51.89±2.21 | 55.42±2.40 | **58.66±2.37** |
| JAFFE | 67.28±6.18 | 66.38±6.15 | 70.26±6.35 | 78.22±4.30 | 69.86±7.89 | 80.80±3.52 | 81.20±3.77 | 80.00±4.49 | 77.96±4.11 | 81.62±2.97 | **83.43±1.90** |
| ATT40 | 51.28±3.86 | 49.16±2.80 | 53.19±3.49 | 55.48±3.14 | 49.56±3.00 | 58.66±2.75 | 55.85±2.89 | 55.18±2.65 | 54.01±2.35 | 55.39±3.05 | **59.31±2.37** |
| TOX-171 | 43.65±3.14 | 41.29±2.48 | 46.90±3.31 | 47.78±1.78 | 43.04±2.51 | 43.77±1.32 | 57.60±0.70 | 53.95±2.98 | 43.10±1.54 | 49.15±3.99 | **58.89±1.30** |
| Isolet | 57.35±3.44 | 58.09±2.72 | 57.73±2.80 | 67.27±2.25 | 61.52±2.97 | 70.72±1.58 | 65.94±2.24 | 60.32±2.38 | 65.24±2.03 | 47.87±2.09 | **71.83±3.31** |
| binalpha | 41.17±1.87 | 39.56±1.51 | 41.66±1.44 | 42.34±1.75 | 39.70±2.08 | 43.82±2.12 | 43.39±1.35 | 43.31±1.72 | 42.64±1.66 | **45.47±1.65** | 45.08±1.36 |
| lung_small | 67.81±7.42 | 67.74±5.56 | 69.93±6.83 | 79.59±5.41 | 65.00±6.49 | 82.26±5.47 | 80.00±5.09 | 78.15±5.29 | 81.92±5.02 | 79.18±5.49 | **83.97±5.55** |
| PCMAC | 50.48±0.50 | 50.63±0.00 | 50.49±0.00 | 56.82±0.00 | 50.54±0.00 | 58.17±0.77 | 51.47±1.07 | 51.94±0.40 | 58.17±0.75 | 56.12±0.21 | **58.84±0.08** |
| RELATHE | 54.45±5.44 | 54.67±0.00 | 54.89±0.70 | 59.06±0.12 | 54.66±0.00 | 57.52±1.56 | 54.53±0.11 | 56.44±0.39 | **61.21±0.75** | 58.85±0.26 | 59.78±0.00 |

**Table 5. Best NMI for different methods on different datasets (mean±std%).**

| Dataset | Baseline | LS | MCFS | SPFS | VSCDFS | AUFS | GLUFS | HSL | LRPFS | RAFG | NFRFS |
|---|---|---|---|---|---|---|---|---|---|---|---|
| Yale | 46.48±2.88 | 44.15±1.67 | 49.23±2.79 | 46.17±2.27 | 44.35±1.92 | 56.34±2.87 | 51.37±1.67 | 57.39±7.79 | 46.23±1.26 | 57.49±2.81 | **59.10±2.15** |
| lung | 60.37±5.38 | 47.04±3.14 | 59.87±5.85 | 65.04±0.88 | 53.47±5.36 | 64.74±1.30 | 66.49±4.01 | 64.36±1.89 | 51.66±0.98 | 65.87±1.34 | **66.78±3.16** |
| COIL20 | 75.58±1.64 | 70.53±1.45 | 74.25±2.04 | 79.73±1.35 | 75.13±1.31 | 79.76±1.02 | 78.65±1.29 | **80.91±1.45** | 78.05±1.29 | 76.82±0.86 | 80.04±1.04 |
| warpPIE10P | 25.36±3.18 | 50.09±3.36 | 30.74±3.57 | 56.41±2.54 | 22.62±2.18 | 59.87±2.10 | 39.78±3.42 | 48.19±2.30 | 26.31±1.31 | 56.87±1.84 | **63.10±1.47** |
| warpAR10P | 20.28±5.42 | 35.23±2.91 | 29.47±2.78 | 51.68±1.67 | 22.05±3.52 | 43.57±1.67 | 42.49±3.60 | 44.77±2.39 | 47.41±3.06 | 44.66±2.73 | **53.18±2.71** |
| ORL | 74.26±1.82 | 63.93±1.46 | 74.81±1.71 | 72.59±1.04 | 71.68±1.32 | 74.76±1.20 | 75.82±1.19 | 75.59±1.59 | 73.10±1.21 | 76.23±1.63 | **78.19±1.05** |
| JAFFE | 73.14±3.55 | 71.27±3.52 | 75.77±3.26 | 82.39±1.68 | 76.68±3.62 | 82.32±2.04 | 82.75±1.78 | 83.26±2.65 | 81.37±1.79 | 83.05±2.63 | **84.67±1.50** |
| ATT40 | 74.02±1.79 | 72.23±1.39 | 75.30±1.56 | 75.94±1.57 | 72.41±1.63 | 78.13±0.95 | 76.32±1.27 | 76.13±0.61 | 73.69±0.87 | 76.20±0.68 | **78.53±1.31** |
| TOX-171 | 15.87±4.44 | 16.44±1.33 | 22.69±4.12 | 23.48±1.15 | 12.19±1.53 | 14.97±0.96 | 30.01±0.77 | 34.52±3.69 | 13.49±1.26 | 34.40±0.88 | **36.96±1.20** |
| Isolet | 75.07±1.71 | 74.14±1.14 | 75.29±1.18 | 77.98±0.87 | 75.86±1.29 | 80.01±1.09 | 78.69±1.07 | 73.99±0.69 | 78.00±0.90 | 64.74±0.82 | **81.58±0.89** |
| binalpha | 57.71±0.87 | 55.85±1.24 | 58.29±0.66 | 57.58±0.67 | 55.25±1.03 | 59.05±0.85 | 58.53±0.76 | 59.19±0.51 | 58.28±0.90 | **60.64±0.76** | 59.68±0.93 |
| lung_small | 65.15±7.16 | 64.70±4.42 | 66.62±5.86 | 74.48±3.34 | 62.72±4.66 | 76.61±4.23 | 74.52±3.14 | 74.30±3.39 | 77.39±5.08 | 73.32±3.01 | **78.36±4.33** |
| PCMAC | 0.04±0.03 | 1.24±1.09 | 0.01±0.00 | 3.09±1.43 | 1.91±0.35 | 4.71±0.00 | 1.34±0.00 | 4.38±0.43 | 2.32±0.00 | 4.69±1.13 | **12.18±0.00** |
| RELATHE | 0.22±0.21 | 0.96±0.78 | 1.03±0.65 | 9.39±0.56 | 1.69±0.76 | 6.52±0.11 | 0.48±0.33 | 4.43±2.71 | 9.16±0.20 | 5.87±1.04 | **9.46±0.01** |

algorithm demonstrates superior performance. In particular, on the lung, warpPIE10P, and TOX-171 datasets, the ACC has been increased by more than 9%, and on the Yale and TOX-171 datasets, the NMI has been increased by more than 12%. This is because NFRFS adopts adaptive graph regularization and utilizes the robust $l_{2,p}$-norm loss function. Consequently, NFRFS has an advantage in feature selection performance.

Figs 2 and 3 show the relationship between the number of features selected by 11 unsupervised clustering methods on 14 datasets and the best accuracy (ACC) and normalized mutual information (NMI). The horizontal axis represents the number of selected features, and the vertical axis represents ACC and NMI. Compared with direct K-means clustering, NFRFS can achieve a relatively high performance even with a smaller number of selected features. Overall, with the increase in the number of selected features, the ACC and NMI curves of NFRFS first ascend and then descend. This is because real-world datasets usually contain some discriminative features and a large number of noisy features. When very few features

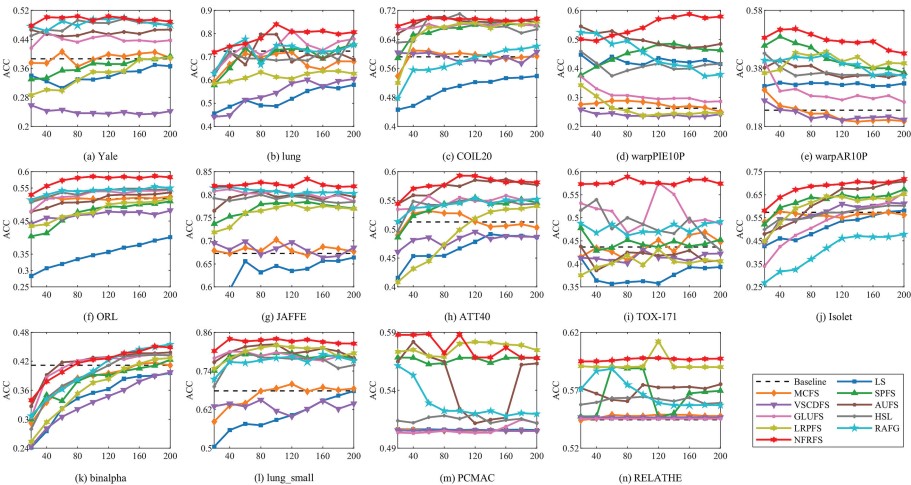

**Fig 2. ACC with different number of features on different datasets.**

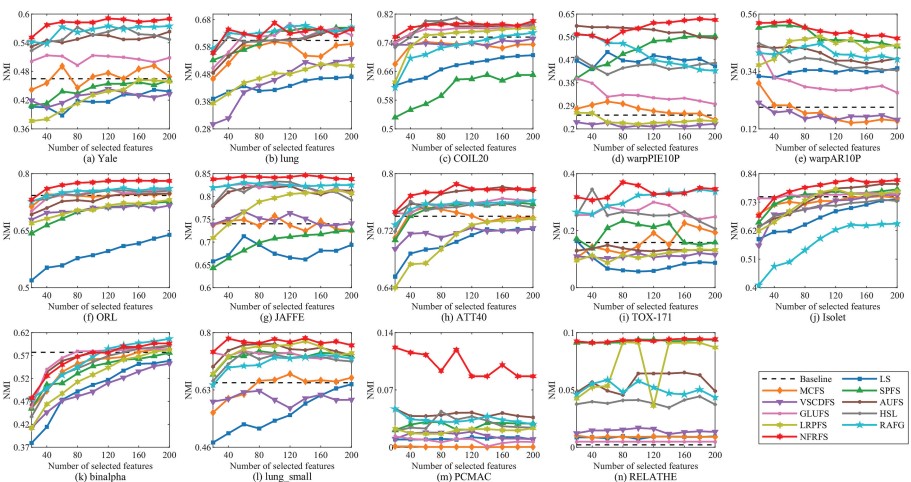

**Fig 3. NMI with different number of features on different datasets.**

are selected, only a part of the discriminative features are excluded. As the number of selected features increases, more discriminative features will be included, which will improve the clustering performance. When further increasing the features, noisy features rather than discriminative features will inevitably be included, thus degrading the clustering performance. In the case of different numbers of selected features, the clustering performance of some algorithms fluctuates significantly, but the NFRFS algorithm changes relatively smoothly, indicating that the number of selected features does not severely affect the clustering effect of NFRFS.

On the Yale, warpAR10P, ORL, JAFFE, TOX-171 and lung_small datasets, the clustering performance of NFRFS corresponding to all numbers of selected features is higher than that of other comparative algorithms. Briefly, regardless of the number of selected features, in most experimental results on all datasets, our proposed method is consistently superior to other state-of-the-art related methods.

## 4.6. Parameter sensitivity analysis

To explore the impact of various parameters on how improving model performance, we choose 8 datasets to experiment a detailed analysis. There are three critical parameters $\alpha$ and $\beta$, and the $l_{2,p}$-norm parameter $p$ to innvolve the adjustment of in NFRFS algorithm. Specifically, $\alpha$ and $\beta$ are adjusted within range $\{10^{-6}, 10^{-4}, 10^{-2}, 1, 10^2, 10^4, 10^6\}$, while $p$ is tuned within range $p = \{0.01, 0.05, 0.1, 0.5, 1\}$. The parameter $\alpha$ controls the weight of manifold learning in the model, $\beta$ regulates the weight of the inner product regularization term in the feature selection matrix, and $p$ reflects the influence of adaptive distance metrics on the model. To isolate the effect of a single parameter, the other two parameters are fixed at 1 during the analysis.

Figs 4 and 5 display the influence of $\alpha$ on ACC and NMI across different datasets under varying numbers of selected features. The results indicate that when fewer features are selected, both ACC and NMI undergo significant changes. However, as the number of selected features increases, these metrics gradually stabilize. This is because a higher $\alpha$ enhances the weight of adaptive graph learning in the objective function, enabling the model to better exploit the intrinsic structure of the selected features and thus guide feature selection

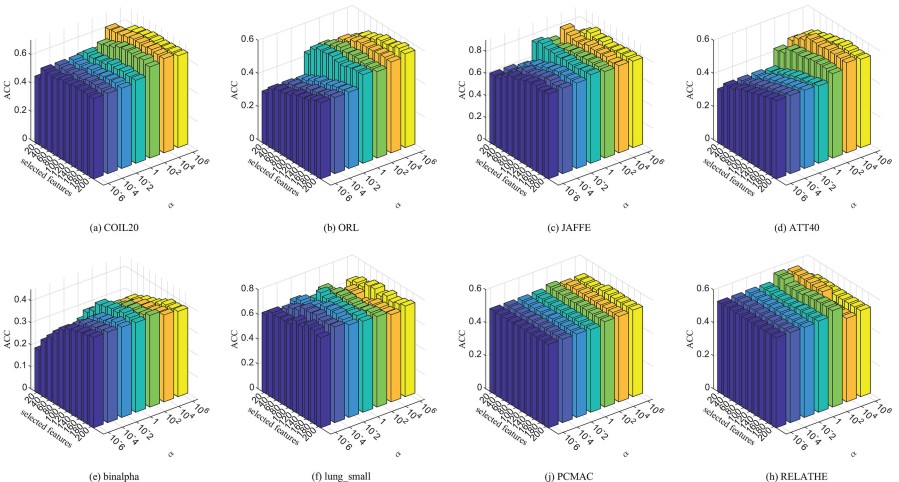

**Fig 4. ACC of NFRFS with different values of $\alpha$ on different datasets.**

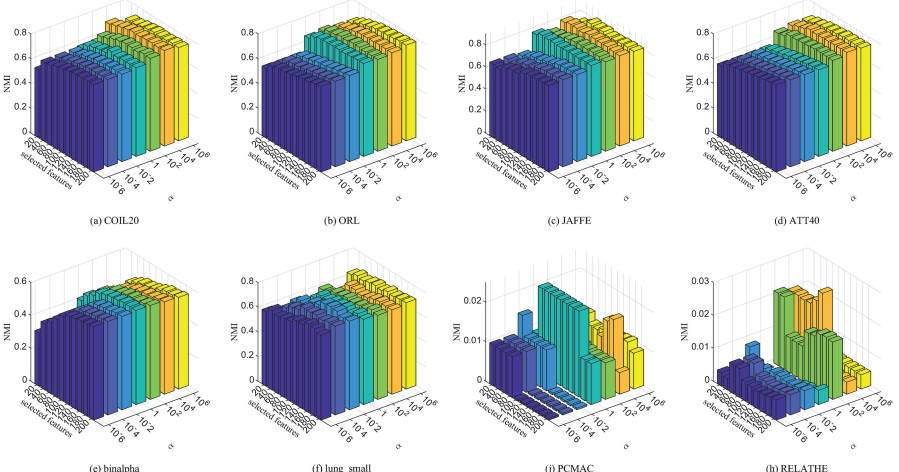

**Fig 5. NMI of NFRFS with different values of $\alpha$ on different datasets.**

more effectively. For the PCMAC and RELATHE datasets, which have relatively low NMI values, changes in $\alpha$ lead to more pronounced impacts on clustering performance. Therefore, it is recommended that $\alpha$ be chosen from the range $\{1, 10^2, 10^4, 10^6\}$.

Figs 6 and 7 plot the effect of $\beta$ on ACC and NMI across different datasets and varying numbers of selected features. The results demonstrate that $\beta$ has a noticeable impact on small datasets, such as lung_small. However, in general, the clustering results are more stable compared to variations in $\alpha$. This finding suggests that the inner product regularization of the feature selection matrix has a relatively minor effect on model performance.

Figs 8 and 9 illustrate the impact of $p$ on ACC and NMI across different datasets and varying numbers of selected features. When $p = 1$, the JAFFE and binalpha datasets achieve the best clustering performance. However, for datasets like ATT40, lung_small, PCMAC, and

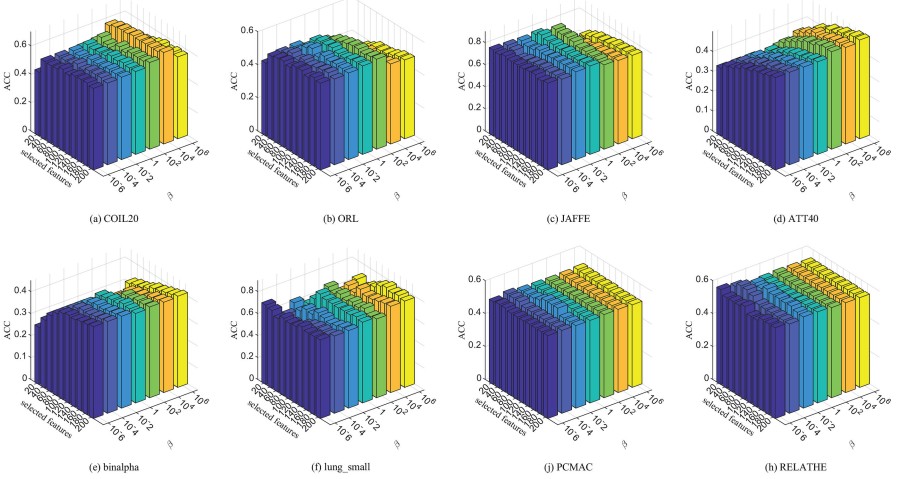

**Fig 6. ACC of NFRFS with different values of $\beta$ on different datasets.**

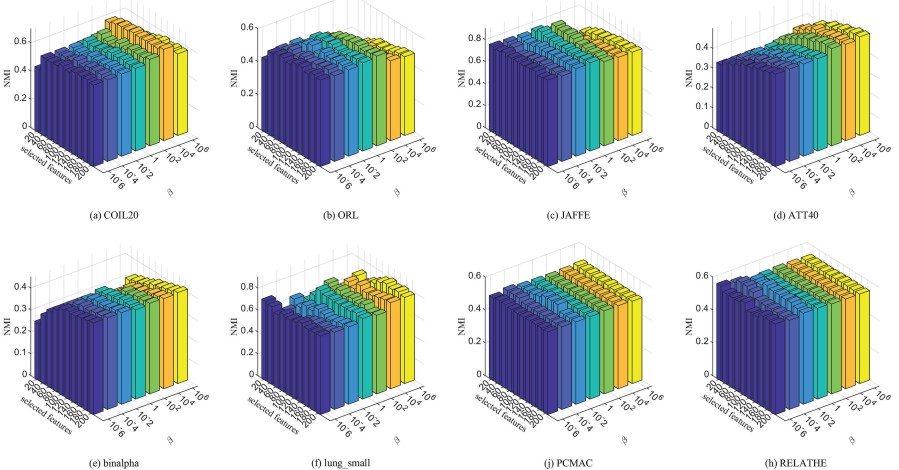

**Fig 7. NMI of NFRFS with different values of $\beta$ on different datasets.**

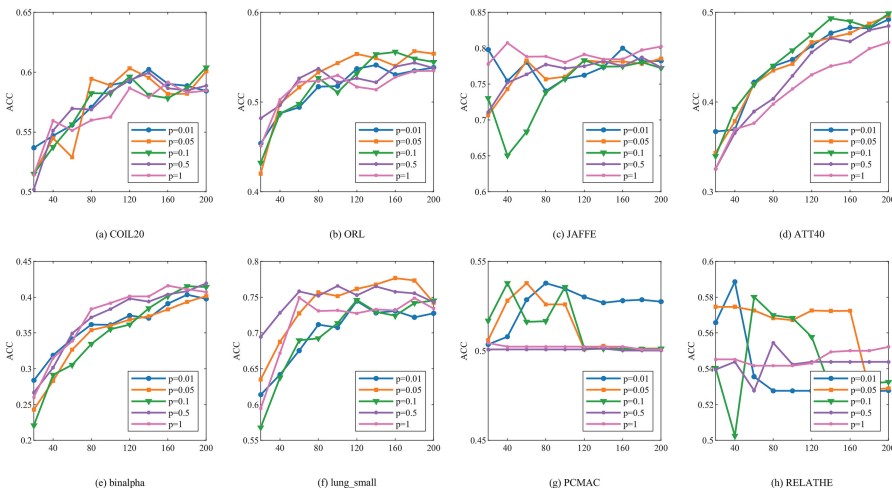

**Fig 8. ACC of NFRFS with different values of $p$ on different datasets.**

RELATHE, clustering results are less favorable. This highlights that fixed distance metrics cannot universally adapt to the feature reconstruction spaces of all datasets. Thus, selecting an appropriate $p$ value based on the specific characteristics of each dataset is essential.

In conclusion, the optimal values of parameters $\alpha$, $\beta$ and $p$ differ across datasets to achieve the best average classification performance. Therefore, in practical applications, these parameters should be flexibly adjusted to obtain optimal results.

## 4.7. Evaluation on robustness

To verify the robustness of NFRFS, noise tests were conducted on the ORL dataset. Each sample was randomly subjected to noise blocks of sizes 8×8, 10×10, and 12×12, as shown in Fig 10. The images in the first row have 8×8 block noise, those in the second row have 10×10 block noise, and those in the third row have 12×12 block noise. Table 6 presents the clustering

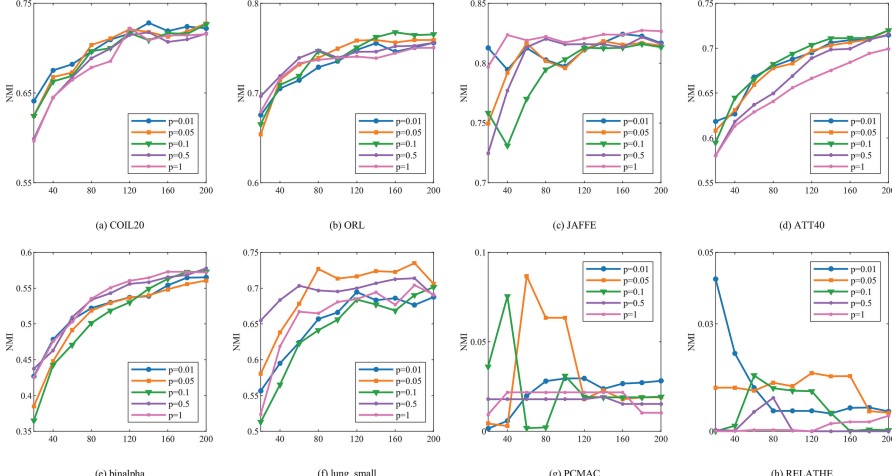

**Fig 9. NMI of NFRFS with different values of *p* on different datasets.**

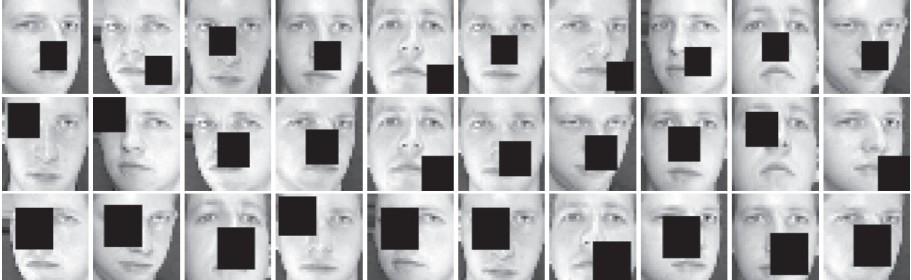

**Fig 10. Representative images of the ORL face dataset with noise block.**

results of various algorithms. The best values for each dataset are highlighted in bold, and the second-best values are underlined. It can be seen that NFRFS outperforms the other methods. Despite the influence of various noise levels, the comparative methods still achieved excellent performance. NFRFS demonstrates strong capabilities in feature recognition and noise reduction.

## 4.8. Ablation study

This section will validate whether adaptive graph learning can enhance the model's clustering performance through ablation experiments. By setting the parameters, the following three effective combinations are obtained:

1. Base Module (*b*): the basic self-expression module which contains only the first and third items of Eq (7),

$$\min \|X - XWH\|_{2,p}^p + \beta \left( Tr(1_{d \times d} WW^T) - Tr(WW^T) \right) \tag{37}$$

2. $b + \alpha + \beta$: the objective function of the baseline and the $\alpha$ weighted item,

$$\min \|X - XWH\|_{2,p}^p + \alpha Tr(W^T X^T LXW) + \beta \left( Tr(1_{d \times d} WW^T) - Tr(WW^T) \right) \tag{38}$$

**Table 6. ACC and NMI for all algorithms on the dataset ORL noise block.**

| Methods | ACC | | | NMI | | |
|---|---|---|---|---|---|---|
| | 8×8 | 10×10 | 12×12 | 8×8 | 10×10 | 12×12 |
| Baseline | 34.96 | 32.71 | 31.81 | 58.57 | 55.75 | 54.48 |
| LS | 36.55 | 36.90 | 36.37 | 60.56 | 60.29 | 59.76 |
| MCFS | 41.09 | 34.20 | 32.38 | 63.64 | 56.49 | 55.91 |
| SPFS | 32.74 | 29.76 | 30.42 | 57.74 | 53.62 | 53.70 |
| VSCDFS | 37.76 | 35.51 | 36.03 | 61.05 | 58.76 | 59.30 |
| AUFS | 43.09 | 39.08 | 38.93 | 65.47 | 63.43 | 63.61 |
| GLUFS | 43.05 | 40.06 | 36.79 | 65.55 | 61.81 | 58.83 |
| HSL | 48.94 | 49.64 | 49.42 | 71.60 | 71.85 | 70.55 |
| LPRFS | 39.59 | 39.47 | 38.86 | 64.45 | 63.39 | 63.41 |
| RAFG | 45.61 | 44.44 | 47.39 | 69.13 | 68.06 | 69.63 |
| NFRFS | **54.04** | **53.49** | **52.96** | **75.12** | **75.22** | **74.17** |

**Table 7. ACC and NMI of the component modules in our model.**

| | | Base Module (b) | $b + \alpha + \beta$ | NFRFS |
|---|---|---|---|---|
| **warpPIE10P** | ACC | 34.71 | 52.55 | 58.74 |
| | NMI | 57.53 | 62.17 | 63.10 |
| **JAFFE** | ACC | 80.19 | 79.79 | 83.43 |
| | NMI | 82.43 | 82.31 | 84.67 |

3. NFRFS: in our proposed method (NFRFS), the objective function of Eq (7).

The clustering results Table 7 show the three combinations on our selected datasets warp-PIE10P and JAFFE under the above three combinations. By comparing the feature selection results before and after introducing adaptive graph learning, the improvement in model performance brought by graph learning is demonstrated. The reason lies in the fact that adaptive graph learning continuously updates the similarity matrix during the feature selection process, focusing on features with strong local correlations while preserving the global structure of the data. This dynamic interaction makes the feature selection process more robust and effective.

## 4.9. Convergence study

Additionally, Fig 11 plots the convergence curves of the NFRFS objective function on 14 datasets. In Fig 11 , the horizontal axis represents the number of iterations and the vertical axis represents the value of the objective function. The results show that the objective function value decreases rapidly and does not increase in subsequent iterations, which indicates that the proposed method is effective and verifies its convergence.

## 5. Conclusion

In this study, we introduce a novel unsupervised feature selection algorithm based on $l_{2,p}$-norm feature reconstruction. This approach adeptly manipulates the distance between the original data space and the reconstructed subspace by adjusting the $l_{2,p}$-norm, thereby enhancing the model's robustness against noise and outliers. The algorithm incorporates inner product sparse regularization to sparsely process the rows and columns of the feature selection matrix, effectively selecting features that are both representative and low in redundancy. Furthermore, the algorithm integrates adaptive structure learning into the objective function

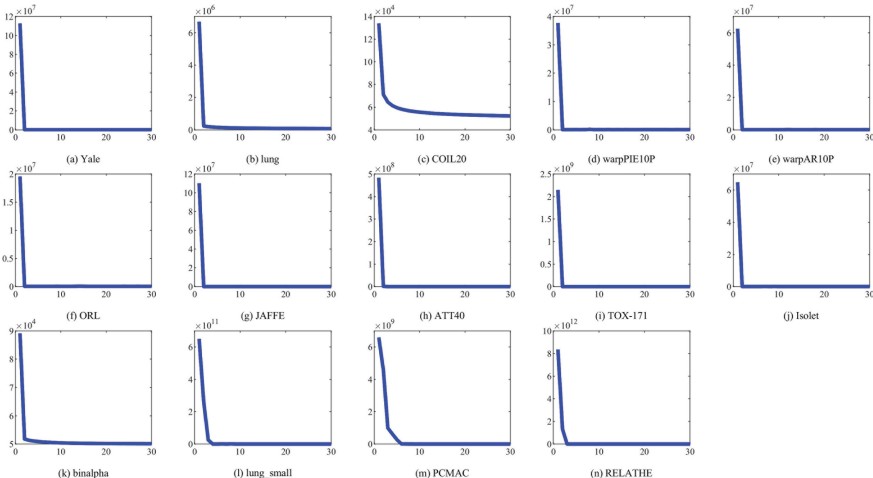

**Fig 11. Convergence curves of NFRFS on different datasets.**

of feature selection to preserve the local structure of the data. Experimental results confirm the superior performance of the NFRFS algorithm in feature selection tasks across various datasets. Nevertheless, determining the optimal hyperparameter $p$ in the feature reconstruction term in a more theoretical and efficient manner remains a topic for future research to explore.

## Supporting information

**S1 Data.**
(ZIP)

## Author contributions

**Conceptualization:** Qian Ning, Guangwei Liu.

**Data curation:** Qian Ning, Yixin Zhu, Miao Zhong.

**Formal analysis:** Qian Ning, Miao Zhong.

**Funding acquisition:** Wei Liu, Guangwei Liu.

**Methodology:** Wei Liu, Qian Ning, Yixin Zhu.

**Software:** Yixin Zhu.

**Supervision:** Haonan Wang.

**Validation:** Qian Ning.

**Visualization:** Miao Zhong.

**Writing – original draft:** Qian Ning, Yixin Zhu.

**Writing – review & editing:** Wei Liu, Guangwei Liu, Haonan Wang, Miao Zhong.

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
