## [Decision Letter · Decision Letter 0]

12 Nov 2024

PONE-D-24-45877Unsupervised Feature Selection Algorithm Based on -Norm Feature ReconstructionPLOS ONE

Dear Dr. Liu,

Thank you for submitting your manuscript to PLOS ONE. After careful consideration, we feel that it has merit but does not fully meet PLOS ONE’s publication criteria as it currently stands. Therefore, we invite you to submit a revised version of the manuscript that addresses the points raised during the review process.

We look forward to receiving your revised manuscript.

Kind regards,

Lei Chu

Academic Editor

PLOS ONE

Journal Requirements:

2. Please note that PLOS ONE has specific guidelines on code sharing for submissions in which author-generated code underpins the findings in the manuscript. In these cases, we expect all author-generated code to be made available without restrictions upon publication of the work. 

Please review our guidelines at https://journals.plos.org/plosone/s/materials-and-software-sharing#loc-sharing-code and ensure that your code is shared in a way that follows best practice and facilitates reproducibility and reuse.

4. Please note that funding information should not appear in the Acknowledgments section or other areas of your manuscript. We will only publish funding information present in the Funding Statement section of the online submission form. Please remove any funding-related text from the manuscript. 

5. Thank you for stating in your Funding Statement: 

“This work was supported in part by the National Natural Science Foundation of China (Grant No.52374123), in part by the Basic Scientific Research Project of the Liaoning Provincial Department of Education (Project No. LJ212410147013, LJ212410147019), and in part by LiaoNing Revitalization Talents Program (Project No.XLYC2211085).”

Please provide an amended statement that declares *all* the funding or sources of support (whether external or internal to your organization) received during this study, as detailed online in our guide for authors at http://journals.plos.org/plosone/s/submit-now.  

Please also include the statement “There was no additional external funding received for this study.” in your updated Funding Statement. 

6. In the online submission form you indicate that your data is not available for proprietary reasons and have provided a contact point for accessing this data. Please note that your current contact point is a co-author on this manuscript. According to our Data Policy, the contact point must not be an author on the manuscript and must be an institutional contact, ideally not an individual. Please revise your data statement to a non-author institutional point of contact, such as a data access or ethics committee, and send this to us via return email. Please also include contact information for the third party organization, and please include the full citation of where the data can be found.

7. When completing the data availability statement of the submission form, you indicated that you will make your data available on acceptance. We strongly recommend all authors decide on a data sharing plan before acceptance, as the process can be lengthy and hold up publication timelines. Please note that, though access restrictions are acceptable now, your entire data will need to be made freely accessible if your manuscript is accepted for publication. This policy applies to all data except where public deposition would breach compliance with the protocol approved by your research ethics board. If you are unable to adhere to our open data policy, please kindly revise your statement to explain your reasoning and we will seek the editor's input on an exemption. Please be assured that, once you have provided your new statement, the assessment of your exemption will not hold up the peer review process.

**Additional Editor Comments:**

The reviewers recognize the potential of the proposed L2,p-norm reconstruction method and its contribution to feature selection. However, after careful evaluation, it has been determined that the manuscript requires significant revisions to address theoretical gaps, enhance experimental rigor, and improve clarity in presentation.

Reviewers' comments:

Reviewer's Responses to Questions

**Comments to the Author**

1. Is the manuscript technically sound, and do the data support the conclusions?

Reviewer #1: Yes

Reviewer #2: Yes

2. Has the statistical analysis been performed appropriately and rigorously? 

Reviewer #1: Yes

Reviewer #2: Yes

3. Have the authors made all data underlying the findings in their manuscript fully available?

Reviewer #1: Yes

Reviewer #2: Yes

4. Is the manuscript presented in an intelligible fashion and written in standard English?

Reviewer #1: Yes

Reviewer #2: Yes

5. Review Comments to the Author

Reviewer #1: Remarks on the technical assessment

The manuscript presents a novel unsupervised feature selection algorithm named NFRFS (Unsupervised Feature Selection Algorithm Based on ℓ2,p-Norm Feature Reconstruction). This method addresses limitations of traditional subspace feature selection approaches that rely on fixed distances to calculate residuals, making them less adaptable to varied datasets and less effective in handling noise and outliers. By utilizing a flexible ℓ2,p-norm, the proposed method enhances adaptability and applicability across different datasets through parameter adjustments. The integration of adaptive graph learning aims to maintain the local geometric structure of the data. The authors demonstrate the effectiveness of NFRFS through numerical studies on 12 benchmark microarray datasets, reporting superior clustering results compared to 8 other unsupervised feature selection algorithms.

In conclusion, in general the paper is appropriate to the journal and the aim is interesting. The paper in general is well written. The authors also presented a comprehensive comparison of their method against existing methods. My specific comments are as follows:

Remarks to the author

(1)This manuscript needs to provide a more detailed explanation of time complexity. For example, in Section 3.4, it only mentions that the dimension of the subspace is less than the number of samples and the dimensionality of the data, but does not explain why the time complexity for updating matrix S is omitted.

(2)Some mathematical notations and terms could be clearer, especially for readers who are not deeply familiar with optimization techniques, such as the definition of the Frobenius norm. If possible, the authors should add a table summarizing frequently-used notations and provide brief explanations for these frequently-used notations.

(4)It is important to note that an explanation should be given when an abbreviation is first proposed. For example, when the abbreviation ‘UFS’ first appeared in this manuscript, it was not stated that it refers to ‘unsupervised feature selection’.

(5)Some minor errors exist, such as ‘eq 8’ on line 169, which should be corrected to ‘eq (8)’.

(6)Several relevant references on Unsupervised Feature Selection Algorithms may need to be cited (10.1109/TKDE.2022.3206881, arXiv:2403.10910).

Reviewer #2: Overall, while this paper presents an interesting approach to unsupervised feature selection by incorporating L2,p-norm feature reconstruction with adaptive graph learning, it suffers from several significant limitations. The theoretical foundation is underdeveloped, the experimental validation is insufficient, and the presentation lacks clarity in crucial technical details. While the core idea has merit, the current manuscript requires substantial revisions to meet publication standards.

Comments:

The theoretical analysis of the proposed L2,p-norm reconstruction method is inadequate. The paper lacks rigorous mathematical proofs for the convergence properties of the optimization algorithm and fails to establish theoretical bounds on its performance. A formal analysis of how different values of p affect the feature selection process is missing.

The experimental evaluation is limited in scope. While testing on 12 microarray datasets provides some evidence of effectiveness, the paper should include experiments on other types of high-dimensional data to demonstrate broader applicability. Additionally, comparisons with more recent state-of-the-art unsupervised feature selection methods are needed.

The parameter sensitivity analysis is insufficient. The paper introduces several important parameters (p in L2,p-norm, regularization coefficients) but does not provide a comprehensive study of their impact on performance. Guidelines for parameter selection in practice are missing.

The computational complexity analysis is absent. Given the multiple components of the algorithm (graph learning, feature reconstruction, regularization), a detailed analysis of time and space complexity is crucial for assessing scalability to large datasets.

The justification for using inner product regularization for feature redundancy reduction is not well-developed. The paper should provide theoretical or empirical evidence showing why this particular form of regularization is more effective than alternatives.

The robustness analysis is lacking. The paper claims improved handling of noise and outliers but does not provide systematic experiments to validate this claim under different noise conditions and outlier scenarios.

The adaptive graph learning component needs better explanation. The interaction between graph learning and feature selection is not clearly described, and the impact of this interaction on the final feature selection results is not thoroughly analyzed.

The presentation quality requires improvement. Many technical details are presented without sufficient explanation or motivation. The notation is sometimes inconsistent, and key algorithms are not described with enough detail for reproduction.

The Literature citation is not adequate, and the related work to machine learning should be discussed

1.Sparse feature selection using hypergraph Laplacian-based semi-supervised discriminant analysis

2.Dual Regularized Unsupervised Feature Selection Based on Matrix Factorization and Minimum Redundancy with application in gene selection

6. PLOS authors have the option to publish the peer review history of their article (what does this mean?). If published, this will include your full peer review and any attached files.

Reviewer #1: No

Reviewer #2: No

---

## [Author Response · Author response to Decision Letter 1]

31 Dec 2024

Thank you to the reviewers for their professional suggestions on our manuscript. For detailed responses, please see the document: Response to Reviewers.pdf

---

## [Decision Letter · Decision Letter 1]

16 Jan 2025

Unsupervised Feature Selection Algorithm Based on L 2,p-Norm Feature Reconstruction

PONE-D-24-45877R1

Dear Dr. Liu,

We’re pleased to inform you that your manuscript has been judged scientifically suitable for publication and will be formally accepted for publication once it meets all outstanding technical requirements.

Kind regards,

Khan Bahadar Khan, Ph.D

Academic Editor

PLOS ONE

Additional Editor Comments (optional):

Reviewers' comments:

Reviewer's Responses to Questions

**Comments to the Author**

1. If the authors have adequately addressed your comments raised in a previous round of review and you feel that this manuscript is now acceptable for publication, you may indicate that here to bypass the “Comments to the Author” section, enter your conflict of interest statement in the “Confidential to Editor” section, and submit your "Accept" recommendation.

Reviewer #1: All comments have been addressed

Reviewer #2: All comments have been addressed

2. Is the manuscript technically sound, and do the data support the conclusions?

Reviewer #1: Partly

Reviewer #2: Yes

3. Has the statistical analysis been performed appropriately and rigorously? 

Reviewer #1: Yes

Reviewer #2: Yes

4. Have the authors made all data underlying the findings in their manuscript fully available?

Reviewer #1: No

Reviewer #2: Yes

5. Is the manuscript presented in an intelligible fashion and written in standard English?

Reviewer #1: Yes

Reviewer #2: Yes

6. Review Comments to the Author

Reviewer #1: (No Response)

Reviewer #2: The author has answered satisfactorily the answers of the previous reviewers. The paper is well-written, and the results are sound. The paper deserves to be published.

7. PLOS authors have the option to publish the peer review history of their article (what does this mean?). If published, this will include your full peer review and any attached files.

Reviewer #1: No

Reviewer #2: No

---

## [Editor Report · Acceptance letter]

PONE-D-24-45877R1

PLOS ONE

Dear Dr. Liu,

I'm pleased to inform you that your manuscript has been deemed suitable for publication in PLOS ONE. Congratulations! Your manuscript is now being handed over to our production team.

Kind regards,

on behalf of

Dr. Khan Bahadar Khan

Academic Editor

PLOS ONE